SOFTWARE

# Epydemix: An open-source Python package for epidemic modeling with integrated approximate Bayesian calibration

Nicolò Gozzi[1,2]*, Matteo Chinazzi[2,3], Jessica T. Davis[2], Corrado Gioannini[1], Luca Rossi[1], Marco Ajelli[4], Nicola Perra[2,5,6], Alessandro Vespignani[1,2]

**1** ISI Foundation, Turin, Italy, **2** Laboratory for the Modeling of Biological and Socio-technical Systems, Northeastern University, Boston, Massachusetts, United States of America, **3** The Roux Institute, Northeastern University, Portland, Maine, United States of America, **4** Laboratory for Computational Epidemiology and Public Health, Department of Epidemiology and Biostatistics, Indiana University School of Public Health, Bloomington, Indiana, United States of America, **5** School of Mathematical Sciences, Queen Mary University of London, London, United Kingdom, **6** The Alan Turing Institute, London, United Kingdom

\* nicolo.gozzi@isi.it

## Abstract

We present Epydemix, an open-source Python package for the development and calibration of stochastic compartmental epidemic models. The framework supports flexible model structures that incorporate demographic information, age-stratified contact matrices, and dynamic public health interventions. A key feature of Epydemix is its integration of Approximate Bayesian Computation (ABC) techniques to perform parameter inference and model calibration through comparison between observed and simulated data. The package offers a range of ABC methods such as simple rejection sampling, simulation-budget-constrained rejection, and Sequential Monte Carlo (ABC-SMC). Epydemix is modular, and supports ABC-based calibration both for models defined within the package and for those developed externally. To demonstrate the computational framework capabilities, we discuss usage examples that include (i) simulating an intervention-driven model with time-varying parameters, and (ii) benchmarking calibration performance using synthetic epidemic data. We further illustrate the use of the package in a retrospective case study that includes scenario projections under alternative intervention assumptions. By lowering the barrier for the implementation of computational and inference approaches, Epydemix makes epidemic modeling more accessible to a wider range of users, from academic researchers to public health professionals.

**Data availability statement:** The data and code to replicate all the presented analyses can be found at https://github.com/epistorm/epydemix/tree/main.

**Funding:** M.A., M.C., J.T.D., and A.V. acknowledge support from the CDC-RFA-FT-23-0069 cooperative agreement from the CDC's Center for Forecasting and Outbreak Analytics. The findings and conclusions in this study are those of the authors and do not necessarily represent the official position of the funding agencies. Any use of trade, firm, or product names is for descriptive purposes only and does not imply endorsement by the U.S. Government. N.G., C.G., and L.R. acknowledge support from the Lagrange Project of the ISI Foundation, funded by Fondazione CRT. The funders had no role in study design, data collection and analysis, decision to publish, or preparation of the manuscript.

# 1 Introduction

Over the last decade, the number of projects aiming to simplify and expand access to epidemic models has steadily grown. The general goal of these efforts is to provide stakeholders (e.g., public health officials, scientists, concerned citizens) with relatively simple and standardized tools to explore, simulate, and implement state-of-the-art epidemic models. The type, target, scale, and flexibility of the proposed solutions reflect the wide heterogeneity of modeling frameworks, diseases, end-users, and applications.

Although a range of epidemic modeling tools exists, built-in support for models' calibration is often missing. This step, which is one of the most critical phases of the modeling pipeline, involves estimating unknown parameters—such as transmission rates, seasonal forcing, or behavioral responses to interventions—by aligning models' outputs with empirical data (e.g., case incidence, hospitalizations, or deaths). Calibration typically requires the definition of a target variable and a corresponding loss function (e.g., mean squared error, mean absolute percentage error), transforming the task into an optimization problem. The goal is to explore the parameter space to identify configurations that minimize the discrepancy between simulated outcomes and observed data. Calibration is essential for fitting epidemic models to data, gaining insights into outbreak dynamics, generating predictions, and evaluating plausible intervention scenarios. However, as with many optimization problems, calibration poses significant challenges. Depending on the model structure and number of parameters to be estimated, the process can be computationally intensive. Moreover, the parameter space is often degenerate, with multiple regions yielding similarly low loss values, complicating the reliable identification of representative parameters. These challenges are further exacerbated by the inherent noise and uncertainty in real-world data, which can further complicate the calibration process [1]. Despite these challenges, calibration remains a necessary step for applying epidemic models to real-world scenarios and informing public health decisions.

Here, we introduce *Epydemix*, an open-source Python package developed to support a broad range of computational tasks in epidemic modeling: from models' specification and simulation to visualization and parameters' calibration. Epydemix supports the incorporation of age structures, contact matrices, and a variety of public health interventions, offering flexibility for modeling complex epidemiological dynamics. A key feature of Epydemix is its built-in support for models' calibration using Approximate Bayesian Computation (ABC) methods [2–4]. The package includes both basic rejection algorithms and more sophisticated approaches such as Sequential Monte Carlo (ABC-SMC) [4]. Epydemix can also be integrated with external modeling efforts. In particular, its calibration routines are designed to be compatible with models developed outside the framework, provided that they conform to the necessary specifications.

In the following sections, we describe the core functionalities and design principles of the Epydemix package. We then illustrate its capabilities through two examples and one detailed case study. The first example demonstrates how Epydemix

can be used to construct a basic epidemic model, integrate country-specific demographic and contact matrix data, and simulate changes in epidemiological parameters and contact patterns resulting from public health interventions. In the second example, we showcase the use of Epydemix's calibration framework by fitting a model to synthetically generated daily incidence data. We apply all three supported ABC algorithms and compare the resulting posterior distributions for the model parameters. Finally, as a case study, we develop a realistic epidemic model to reproduce reported, weekly COVID-19 deaths in Massachusetts during the early months of 2020. The calibrated model is then used to evaluate the potential impact of various non-pharmaceutical intervention strategies by simulating changes in contact rates over the summer of 2020 and estimating their effect on projected COVID-19 deaths under different scenarios.

Overall, this work contributes to ongoing efforts to streamline and broaden access to advanced epidemic modeling. By lowering technical barriers, the package aims to make state-of-the-art methodologies in epidemiological modeling more accessible to researchers, public health practitioners, and policy analysts alike.

## 1.1 Related work

Among the computational approaches made available to the public we find software tools that facilitate exploring and/or running epidemic models via graphical user interfaces (GUI), applications programming interfaces (API), or Web applications [5–8]. These include open-source simulators, written in Python, Julia, R, C++, and other programming languages, offering complete, often adaptable, implementations of different classes of epidemic models [9–24] and software libraries (e.g., Python or R packages) that act as building blocks for abstract and personalized implementations [8,13,19,25–30]. In terms of the analytical framework at the basis of these projects, we find compartmental [6,11], metapopulation [5], agent-based [8,10,12,13,16,17,19,24,24,25], and network-based [9,15,18,20,21,28,29] models. We also find projects offering a range of different models under the same umbrella [7,14,23,26,27,30]. While some tools target a specific disease such as COVID-19 [6,8,11,16,24], HIV [20], or influenza [12], others focus on classes of infectious diseases linked to specific types of transmission dynamics [5,9,10,13–15,19,25,27,29,30]. There is also heterogeneity in terms of the spatial scale of the analysis that these tools allow: from close contacts within a given area [7,9,15,18,21,22,28–30] to city [16,19,24], country [6,11,12] and global scales [5]. Furthermore, some projects are flexible and allow, within given constraints, the user to set the target scale of geographical analysis as input [8,10,12–14,17,19,20,23,25–27]. For this reason, before moving forward, we briefly situate Epydemix within the landscape of epidemic-modeling software.

Compartmental platforms range from the COVID-focused *COVID-19 Scenarios* [6] (interactive exploration without native calibration) and *SimCOVID* [11] (Simulink/MATLAB with calibration) to metapopulation tools such as *GLEAMviz* [5] (global mobility–aware GUI). In general, these tools emphasize scenario analysis over end-to-end inference such as the one offered in Epydemix.

Another group of tools is defined by agent-based modeling (ABM) approaches. Among those we find *COVASIM* [8] (Python, intra-host features, user-driven fitting), *Agents.jl* [10] (Julia, grid/graph/OSM with parameter search), *Sampy* [25] (Python, modular ABM for stochastic epidemic simulations with spatial structure and intervention modeling), the influenza models *FluTE* [12] and *FRED* [13] (both working on synthetic US populations), urban SARS-CoV-2 tools such as *COMOKIT* [16] and *OpenABM-Covid19* [24], and behavior-centric frameworks like *BESSIE* [17] and *Pyfectious* [19] (the latter with Reinforcement Learning for policy optimization). For this class of models calibration is generally not supported and externalized to other packages.

Network-based tools includes *CSonNet, VTES* (didactic), *GEMFsim* (multi-language, multilayer), *SimpactCyan* (dynamic networks for HIV), *Epinet* (supporting ERGM inference), *EoN* (Python package designed for studying disease spread in static networks), and fast temporal-network SIR implementations [9,15,18,20,21,28,29]. While these tools provides a wide range of options to include network topology in the modeling they generally do not offer extensive inference or calibration features.

Finally, we find tools that allow the implementation of a range of model types such as *EpiFire, MEmilio, Eir, Epilearn* (with ML/statistical forecasting), and *EpiModel* (including deterministic compartmental models, stochastic individual contact models, and stochastic network models) [7,14,23,27,30], that prioritize breadth and the implementation of various models rather than their calibration to real data. In the S1 Text we provide a table with the main characteristics of all software tools reviewed here.

In this context, Epydemix deliberately centers on structured compartmental models, offering native, end-to-end calibration capabilities that are uncommon across existing platforms. It offers a flexible, and modular architecture for rapid model specification, simulation, parameter estimation, and scenario analysis geared to real-time epidemic assessment, policy-oriented planning, and short-term forecasting, especially when data are sparse, noisy, or insufficient to parameterize large-scale ABMs [31,32]. Epydemix is also designed to lower technical barriers by adopting Approximate Bayesian Computation (ABC), a likelihood-free inference framework that enables parameter estimation based solely on model simulations and observed data. This approach has gained traction in the epidemic modeling community in recent years [2,3, 33]. Among other inference methods, ABC provides a conceptually transparent and general framework that directly links simulations to data through summary statistics, without requiring explicit likelihood evaluation. It is particularly suitable for stochastic or individual-based epidemic models, where the probabilistic nature of transmission and observation processes makes likelihood derivation impractical. Furthermore this calibration is general enough to support metapopulation or network models, while preserving reproducibility, interpretability, and a clear separation between model structure, data, and inference process.

## 2 Design and implementation

Epydemix is implemented in the Python programming language. It can be installed from the Python Package Index (PyPI), the standard repository for Python packages, using the command `pip install epydemix` in a terminal or command prompt—ideally within a virtual environment to avoid dependency conflicts. The source code, data, and tutorials are open source and freely available on GitHub [34,35]. The full documentation is available on Read the Docs, a commonly used free software documentation hosting platform [36].

The typical workflow in Epydemix proceeds through several stages (see Fig 1). First, the user defines an epidemic model by specifying the compartmental structure, transition dynamics, and associated parameters. The model can then be linked to real-world demographic data by incorporating population pyramids and age-stratified contact matrices. Temporal variations in parameters and contact patterns—such as those induced by public health interventions—can also be specified to reflect dynamic changes in transmission conditions. Once configured with the initial conditions, the model is simulated over a user-defined time period. Calibration on empirical data is then performed using built-in ABC algorithms, which require observed time series and prior distributions for the parameters of interest. Calibrated models can then be used to generate scenario projections, forecasts, or explore the posterior distributions of free parameters. At each step of the workflow, Epydemix provides visualization tools for examining simulation outputs, demographic structures, and calibration results, including posterior parameter distributions.

In the following sections, we explain in more detail each of these steps. Classes and functions are denoted with a mono-spaced font (i.e., `Class`, `function`). Classes start with a capital letter, while functions with a lowercase letter.

### 2.1 Model definition and stochastic simulations

Epydemix supports the construction of general single-population stochastic compartmental models [37], in which individuals are grouped into compartments according to their epidemiological status (e.g., susceptible, infectious, recovered). Beyond the health status, compartments can also represent other relevant characteristics, such as vaccination status or behavioral attributes associated with the adoption or relaxation of NPIs. The progression of individuals through the different model compartments is represented by stochastic transitions between compartments. These transitions can be

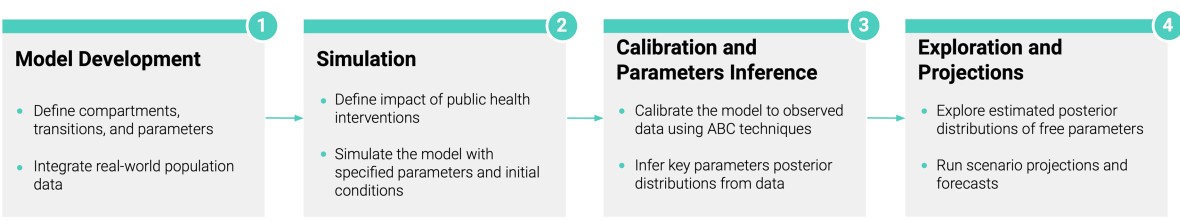

**Fig 1**. **Workflow of a typical Epydemix run.**

broadly classified into two categories: spontaneous transitions, which occur independently of interactions with other individuals (e.g., recovery or waning of immunity), and mediated transitions, which are driven by interactions between individuals in different compartments (e.g., infection due to contact between susceptible and infectious individuals).

For example, in the classic Susceptible-Infected-Recovered (SIR) model [37], susceptible individuals, in compartment $S$, are susceptible to the infection. By contacting infectious individuals, susceptibles may then transition to the infectious stage, and join compartment $I$. The infection process can be described using a notation borrowed from mass action reaction systems: $S + I \xrightarrow{\beta} 2I$, where $\beta$ is the transmissibility of the pathogen. After the typical infectious period, infectious individuals spontaneously transition to the recovered compartment $R$. These transition can be described as $I \xrightarrow{\mu} R$, where $\mu$ is the recovery rate. The infection process is a classic example of a mediated transition. The recovery process instead is an example of a spontaneous transition. The package provides full flexibility in defining both the number and type of compartments, enabling the development of advanced models beyond classic SIR-like structures. As mentioned above, this includes, for example, compartments for vaccinated individuals and behavioral dynamics. Individuals can be further categorized into demographic groups, such as age brackets (see Sect 2.2 for more details). Accordingly, we denote the number of individuals in compartment $X$ (e.g., $X \in \{S, I, R\}$) and demographic group $k$ as $X_k$. Moreover, we define $X_{total}$ as the sum of individuals in compartment $X$ across all demographic groups (i.e., $X_{total} = \sum_{k=1}^{K} X_k$). It follows that, for example, $S_k$ indicates the number of susceptible individuals in demographic group $k$, and $S_{total}$ the total number of susceptible individuals across all groups.

Models are built using the `EpiModel` class which requires: 1) the compartments, defining the possible states of individuals, 2) the transitions among them, and 3) the parameters governing such transitions. Canonical models such as the Susceptible-Infectious-Recovered (SIR), Susceptible-Exposed-Infectious-Recovered (SEIR), and Susceptible-Infectious-Susceptible (SIS) are provided as predefined templates.

Simulations in Epydemix are stochastic and follow a chain binomial approach to model transitions between compartments. At each time step, the number of individuals transitioning from compartment $X_k$ to $Y_k$ is drawn from a binomial distribution, $\text{Bin}(X_k, p_{X_k \to Y_k})$, where $X_k$ denotes the number of individuals currently in the source compartment, and $p_{X_k \to Y_k}$ represents the probability of transition within the interval $\delta t$. By default, transition probabilities are computed by converting transition rates (i.e., $r_{X_k \to Y_k}$) into discrete-time risks using the standard exponential transformation:

$$p_{X_k \to Y_k} = 1 - e^{-r_{X_k \to Y_k} \delta t}$$

When a compartment has multiple possible destination states, transitions are modeled using a multinomial extension of the chain binomial process (see S1 Text).

By default, two types of transitions are supported: spontaneous and mediated transitions. For spontaneous transitions, the rate $r_{X_k \to Y_k}$ is directly defined by the user-specified parameter. In the case of mediated transitions, the rate additionally

incorporates an interaction term form that depends on the size of the mediating compartment (e.g., the number of infectious individuals in transmission processes). For illustrative purposes, let us consider a classic SIR model. In this case, the rate of recovery will be $r_{I_k \to R_k} = \mu_k$, where $\mu_k$ is the inverse of the infectious period. In the simplest case, $\mu_k = \mu$ for all $k$. The rate of infection for group $k$ will be instead $r_{S_k \to I_k} = \beta \sum_{k'} C_{kk'} \frac{I_{k'}}{N_{k'}}$, where $\beta$ is the transmission rate, **C** is the contact matrix (see Sect 2.2 for more details), and $N_{k'}$ is the total number of individuals in group $k'$. We refer the reader to the S1 Text for a more detailed mathematical formulation of the SIR model.

Single stochastic realizations of the model are generated using the `simulate` function, which returns a `Trajectory` object containing the compartment dynamics (e.g., $X_k(t)$), the transition counts (e.g., $X_k(t) \to Y_k(t)$), and the parameters used in the simulations (e.g., the transition rates). The `run_simulations` method performs multiple realizations, returning a `SimulationResults` object with all trajectories and summary statistics like medians and confidence intervals. In Sect 3.1.1 of the Results, we provide a practical example illustrating model definition and simulation.

## 2.2 Integration of real-world population demographic and contact patterns data

By default, the `EpiModel` class is initialized with a population of 100,000 individuals in a single demographic group, equivalent to a homogeneous mixing assumption [37]. However, the `Population` class can be used to define the essential information about the population, including the total number of individuals, their distribution across demographic groups, and the contact rates between groups. This information serves as input for the simulations, determining the initial conditions and the structure of interactions that drive the transmission dynamics. In particular, the demographic composition informs how individuals are allocated at initialization, while contact matrices are typically used to modulate the infection rate between groups during the stochastic simulation process. `Population` objects can also be instantiated using real-world demographic data, allowing users to ground their simulations in realistic settings. Indeed, the package is complemented by an external online repository (*Epydemix Data*), which provides access to population structures and synthetic age-stratified contact matrices for over 400 regions worldwide. The data is hosted in a dedicated GitHub repository [38], and include age-stratified population distributions along with contact matrices that characterize interactions across different settings, including households, workplaces, schools, and community environments. The contact matrices are provided in three distinct variants derived from the synthetic datasets developed by Mistry et al., 2021 [39], Prem et al., 2021 [40], and Prem et al., 2017 [41]. The dataset covers the majority of countries worldwide, with sub-national demographic information available for selected regions. The majority of population data comes from the United Nations World Population Prospects [42]. A full list of supported regions, datasets, and sources is available on GitHub [38].

Users can easily import data using the `load_epydemix_population` function, which creates a `Population` object for the desired location. Alternatively, users can define custom populations by providing their own dataset as input. The resulting `Population` object can be set for simulations with the `set_population` method of `EpiModel`. Additional functionalities simplify population customization and exploration. For example, users can define custom age groups and pass them to `load_epydemix_population` to achieve the desired granularity.

## 2.3 Advanced modeling features

The previous subsections introduced the components required to define and simulate epidemic models. Building on this foundation, the package also supports a range of advanced features that extend its modeling capabilities.

**2.3.1 Defining transition mechanisms.** As previously discussed, the package supports two primary types of transitions: spontaneous and mediated. However, our framework allows users to define custom transition types to accommodate specific modeling requirements. For example, users may want to introduce infection processes where the force of infection differs from the conventional mass-action formulation, or model behavioral dynamics in which transition rates depend non-linearly on the number of individuals in specific states. This can be achieved using the `register_transition_kind` method of the `EpiModel` class, which takes as input a user-defined name for the new transition

type and a function that specifies how to compute its transition rate. Once registered, transitions of this new type can be incorporated into the model. This feature enables the implementation of alternative formulations broadening the range of dynamical processes that can be represented within the framework.

**2.3.2 Time-varying and group-specific parameters.** In the basic model formulation, transition parameters are assumed to be constant. However, Epydemix allows the definition of both time-dependent and group-specific parameters. Specifically, the framework supports: (i) time-varying parameters, such as seasonally modulated transmission rates, which are specified as arrays of length $T$, corresponding to the number of simulation steps; and (ii) group-specific parameters, such as age-dependent susceptibility or recovery rates, represented as arrays with shape $(1,K)$, where $K$ denotes the number of population groups. These two dimensions can be combined to define parameters that vary simultaneously across time and groups, using arrays of shape $(T,K)$. Additionally, transition rates can be expressed symbolically to capture functional relationships between parameters. For example, the transmissibility of a second strain, $\beta_2$, may be defined as a multiple of the baseline rate, such that $\beta_2 = \psi * \beta_1$, where $\psi$ denotes the relative transmissibility. Additional practical examples of parameter specification are provided in the online tutorials [35].

**2.3.3 Non-pharmaceutical interventions and behavioral changes.** In Epydemix, mitigation strategies, including NPIs such as school and workplace closures, can be incorporated into models using two core functionalities of the `EpiModel` class: `add_interventions` and `override_parameter`. The `add_interventions` functionality allows users to introduce time-specific modifications to selected layers of the contact matrix. Interventions can be specified either by (i) applying a uniform reduction factor across all elements of a target contact matrix, or (ii) supplying a custom contact matrix that adjusts both the overall intensity and the relative distribution of contacts between population groups. In both cases, users can define the start and end date of the intervention. This flexible design enables to model targeted interventions— for instance, simulating school or workplace closures that selectively reduce contacts in those settings over a specified period while preserving (or increasing) interactions in others. The `override_parameter` functionality provides a general mechanism for introducing time-dependent modifications to transition parameters during specific periods of the simulation. This allows users to capture dynamic changes in transmission dynamics, such as the combined effects of multiple mitigation measures represented by a temporary reduction in transmissibility. Additionally, `override_parameter` can be employed to model intrinsic temporal variations of the parameters, such as seasonal fluctuations in transmissibility or behavioral shifts over time.

In addition to externally imposed interventions, Epydemix also supports the modeling of self-initiated behavioral changes such as the voluntary avoidance of public spaces, workplaces, or schools, and the adoption of preventive measures like mask-wearing, which have been widely documented during epidemics [43,44]. These dynamics can be represented by introducing specific compartments that capture subpopulations adopting precautionary behaviors that reduce the risk of infection [45]. This compartment-based approach complements the use of dynamic parameters and interventions.

## 2.4 Model calibration

Epydemix supports the full pipeline of epidemic modeling, including parameter inference through simulation-based calibration. Epydemix adopts an Approximate Bayesian Computation approach, which is particularly well suited for stochastic simulation settings, as it enables inference by comparing observed and simulated data without requiring explicit likelihood evaluation [3,4]. The core object for models' calibration is the `ABCSampler`, which requires the following key inputs: i) prior distributions $\pi(\theta)$ for free model parameters $\theta$, ii) a distance function $d(\cdot)$, iii) a simulation function, and iv) the observed data $\mathbf{y}_{obs}$. Prior distributions represent initial beliefs about parameter values and must be specified using `scipy.stats` distribution functions, which offer a wide range of options for both continuous and discrete parameters. For instance, to reflect minimal prior knowledge, one may adopt flat or broad priors, such as a uniform distribution defined over a plausible range or a wide normal distribution centered around an initial guess. The distance function quantifies

the discrepancy between simulated and observed data, guiding parameters' selection. For example, one may use the Euclidean distance between simulated and observed incidence time series, or compute the (absolute) differences on key summary statistics such as peak incidence or epidemic duration. The simulation function is used by the calibration algorithm to generate a model output $\mathbf{y}_i$ and must conform to minimal interface requirements. At its simplest, this function acts as a wrapper around the Epydemix `simulate` function which runs a single stochastic realization for a given model, specifying the output used for distance calculations and parameters' selection. If needed, users can extend the wrapper to include more complex input pre-processing (e.g., custom model initialization) or output transformations (e.g., aggregating compartments or computing summary statistics before distance calculation). This design allows calibration not only of models built with Epydemix but also of external models. As long as the simulation function adheres to the minimal required output structure, models developed outside the package can be used within an `ABCSampler` object. Finally, the observed data represent the empirical measurements (e.g., observed incidence time series) against which model outputs are compared and are required for the calculation of distances during the calibration process.

The package supports three ABC algorithms for parameter inference [2,3]. They can be executed using the `calibrate` method of an `ABCSampler` object, specifying the desired `strategy`.

The basic ABC rejection algorithm is performed by setting `strategy = "rejection"` when calling the `calibrate` method of `ABCSampler`. It requires defining a tolerance $\epsilon$ and a population size $P$. The algorithm iteratively samples the parameters $\theta_i$ from the prior distributions and executes the simulation function to produce an output $\mathbf{y}_i$ which is compared to observed data $\mathbf{y}_{obs}$ using $d(\mathbf{y}_i, \mathbf{y}_{obs})$. Parameters are accepted if $d(\mathbf{y}_i, \mathbf{y}_{obs}) < \epsilon$. This process continues until $P$ parameter sets are accepted, approximating the posterior distribution of free parameters. This approach is simple and intuitive, but has limitations, such as determining an appropriate tolerance $\epsilon$. Smaller values improve accuracy but slow down the calibration, while larger values speed up convergence at the cost of precision. Additionally, the prior distribution remains fixed, ignoring the insights gained during the process.

To address these issues, the package offers an alternative algorithm that can be used by setting `strategy = "top_fraction"`. This replaces the fixed tolerance with a total simulation budget $B$ (i.e., total number of simulations) and a selection percentage $x$. Sampling, simulations, and comparison with data continue until the budget is exhausted, after which the top $x\%$ of parameter sets are selected based on the distance metric. Hence, $\epsilon = Q_{x/100}(d(\mathbf{y}_i, \mathbf{y}_{obs}))$, where $Q_\alpha$ is the $\alpha$-th percentile. This approach is equivalent to set a tolerance implicitly defined by the largest distance for top $x\%$ simulations given the simulation budget.

A more advanced algorithm is an ABC method based on Sequential Monte Carlo (ABC-SMC) that extends the rejection approach by using $T$ generations with progressively smaller tolerances. This calibration method is available by setting `strategy = "smc"`. The implementation of the ABC-SMC methodology follows Ref [4]. In brief, each generation's prior distribution is the posterior from the previous generation, perturbed by a kernel function. This iterative process starts with a broad prior and high tolerance, refining the parameter space with each generation. The final generation yields a refined approximation of the posterior distribution. In the package, by default, the first generation begins with an infinite tolerance, and subsequent values are computed as the median (or another user-specified quantile) of distances from accepted particles in the previous generation [46,47]. Alternatively, a specific tolerance schedule can be provided. A component-wise Gaussian kernel is used for continuous parameters, while discrete parameters employ a discrete jump kernel [48]. Users have the flexibility to define their own perturbation kernels via the `Perturbation` class.

The ABC-SMC algorithm offers the highest accuracy, but can be more computationally demanding. The simple rejection algorithm is a quick and easy option for exploring models or prior distributions. The modified rejection approach ensures results within a fixed time frame, but is generally less accurate. It is particularly useful for exploratory tasks or recurrent processes with strict runtime constraints. For all algorithms, stopping conditions such as maximum runtime, simulation budget, or minimum tolerance (ABC-SMC only) can be defined by the user. The pseudocode for the three calibration algorithms is provided in the S1 Text. For all the three algorithms, the `calibrate` method returns a `CalibrationResults` object, providing access to estimated posterior distributions for free parameters, selected trajectories, and

general information about the parameters used in the calibration algorithm. After the calibration step is completed, the `run_projections` method of `ABCSampler` allows running projections by sampling from the approximated posterior distribution. See Sect 3.1.2 of the Results for a practical example illustrating the model calibration process.

### 2.5 Visualization

The package provides several options to visualize models' features and outputs. Population distributions across demographic groups and contact matrices can be visualized using the `plot_population` and `plot_contact_matrix` functions. The `plot_spectral_radius` function plots the spectral radius of the contact matrix over time. This quantity is proportional to the model's reproduction number [49]. Variations to the spectral radius induced by mitigation policies can be used to quantify the potential impact of interventions on epidemic dynamics. Simulation outputs, such as the median and confidence intervals of the number of individuals across each compartment as well as outputs from the calibration algorithms can also be explored with built-in visualizations. The `plot_quantiles` function displays summary statistics of simulated trajectories, while `plot_posterior_distribution_2d`, `plot_posterior_distribution`, and `plot_distance_distribution` visualize the joint posterior distribution of two parameters, the marginal posterior distribution of a single parameter, and the distribution of distances for accepted parameters, respectively.

## 3 Results

In this section, we first present concrete examples that demonstrate the use of Epydemix across the key stages of epidemic modeling. We then illustrate the application of the package to real-world scenarios through a detailed case study. All code and data necessary to reproduce the examples and case study are publicly available on the Epydemix GitHub repository [35].

### 3.1 Example usages

#### 3.1.1 Model definition and simulation.
We begin by providing an example of how to use the package to define and simulate epidemic models. We consider a simple SIR model, which can be created with the following code:

```python
from epydemix import EpiModel
from epydemix.population import load_epydemix_population

# Define the SIR model
sir_model = EpiModel(
    name="SIR Model",
    compartments=["S", "I", "R"],
    parameters={"beta": 0.045, "mu": 0.1},
)

# Define the transitions
sir_model.add_transition("S", "I", params=("beta", "I"), kind="mediated")
sir_model.add_transition("I", "R", params="mu", kind="spontaneous")

# Import and set population
population = load_epydemix_population("Italy")
sir_model.set_population(population)
```

The first block initializes the model by specifying the compartments and parameters. Both can also be added after initialization using the `add_compartments` and `add_parameter` methods, respectively. Parameters can also be passed directly to transitions as values. The next two blocks define the model's transitions. In both cases, we specify the source and target compartments, the parameters for the transition, and the transition type. For the recovery process (a spontaneous transition), only the transition rate is required, whereas for the infection process, the name of the compartment mediating the transition must also be provided. The final block imports population data for Italy and assigns it to the

model. After these initializations, we can run a given number of stochastic simulations over a defined time window and visualize some outputs by using the following set of instructions:

```
1  # Run stochastic simulations
2  sir_results = sir_model.run_simulations(
3      start_date="2024-01-01",
4      end_date="2024-08-31",
5      Nsim=100,
6  )
7
8  # Plot S, I, R evolution (median, 95% CI)
9  df_quantiles = sir_results.get_quantiles_compartments()
10 plot_quantiles(df_quantiles, columns=["I_total", "S_total", "R_total"])
```

We note how the parameter `Nsim` can be used to set the number of simulations. Unless specified, its default value is set to 100. The output of the code can be seen in Fig 2A, which shows the median and 90% confidence intervals of the number of individuals in compartments $S$, $I$, and $R$ summing them across all age-groups, i.e., $X_{total}$ with $X \in [S, I, R]$.

Next, we incorporate public health interventions to simulate a scenario in which modelers aim to quantify the potential effects of NPIs on an outbreak. The first intervention is a partial school closure, reducing contacts in the school layer to 35% of the original value (i.e., a 65% reduction). This can be added to the model as follows:

```
1  # School closure
2  sir_model.add_intervention(
3      layer_name="school",
4      start_date="2024-03-01",
5      end_date="2024-05-01",
6      reduction_factor=0.35,
7      name="school closure"
8  )
```

The second intervention is a partial workplace closure, reducing contacts in the workplace layer by 70%, which can be done similarly (code not shown). Additionally, we consider a range of other measures, such as mandatory masking and general social distancing policies, which reduce the transmission parameter ($\beta$) over a specified period from 0.045 (the initial value) to 0.02. This measure can be introduced in the model with the following code:

```
1  # Mask/Social distancing
2  sir_model.override_parameter(
3      start_date="2024-02-01",
4      end_date="2024-08-31",
5      name="beta",
6      value=0.02
7  )
```

We finally re-run the model with these interventions and compare the results. Fig 2B shows the output of the `plot_quantiles` function providing the evolution of the total number of infected individuals over time for the SIR model we defined above, and the analogous model without interventions, highlighting the flattening of the epidemic curve due to interventions. Fig 2C visualizes the impact of interventions on contact patterns, plotting the percentage reduction in the spectral radius of the overall contact matrix over time. As mentioned, the spectral radius is the largest eigenvalue of the contact matrix and is proportional to the basic reproduction number [49]. Finally, Fig 2D demonstrates the effects of interventions in terms of total averted infections and the percentage reduction in peak intensity, computed with minimal additional manipulation of the `SimulationResults` outputs.

**3.1.2 Model calibration.** In this second example, we demonstrate how Epydemix can be used to calibrate epidemic models to observed data using ABC techniques. As target data we use synthetically generated daily new infection counts (i.e., daily incidence) from an SIR model applied to a population representative of Indonesia. The synthetic data is obtained by fixing the transition rates to specific values and adding noise to the resulting daily incidence to mimic the variability typical of real data. In particular, we set the transmissibility to 0.02 and the recovery rate to 0.2 days$^{-1}$. At each

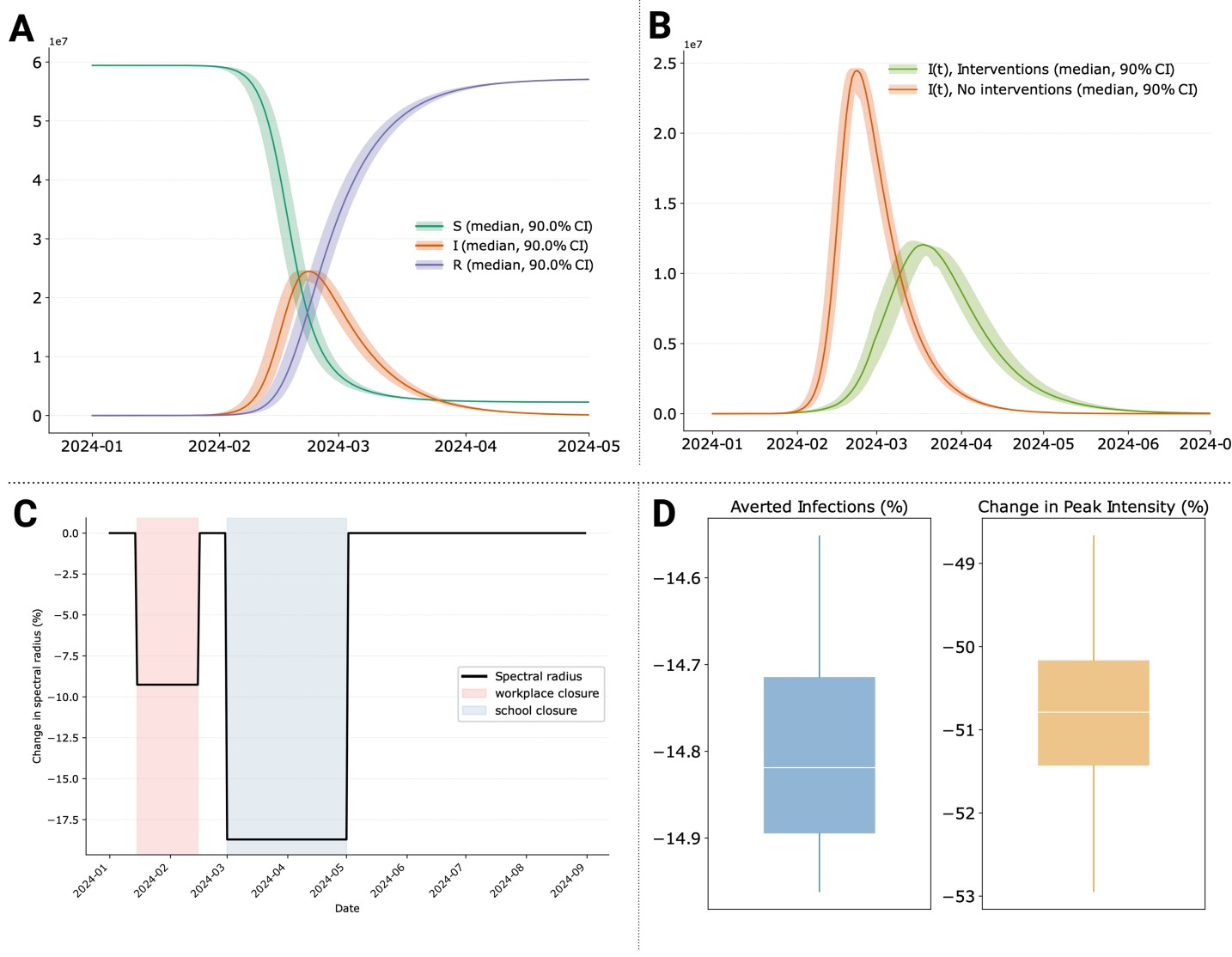

**Fig 2**. **A) Time series showing the simulated number of susceptible, infected, and recovered individuals across all age-groups (median and 90% confidence intervals). B)** Time series showing the total number of infected individuals (median and 90% confidence intervals) with and without interventions, highlighting the impact of NPIs in flattening the epidemic curve. **C)** Percentage change in the spectral radius of the overall contact matrix during workplace and school closures, illustrating the reduction in contact dynamics. **D)** Boxplots summarizing the impact of interventions, including the percentage of averted infections and the reduction in peak size.

time step, we add random noise of up to $\pm 20\%$ to the simulated daily incidence, ensuring that no negative values occur by setting any of such cases to zero.

Following a similar procedure to the one described above, we define a new SIR model which will fit to the synthetic data. In particular, our objective is to estimate the posterior distribution of both the transmission and recovery rates based on the observed data. As a first step, we need to define the simulation function to ensure that the calibration algorithm correctly processes the model's output. This can be achieved as follows:

```
1  from epydemix import simulate
2
3  def simulate_wrapper(parameters):
4      results = simulate(**parameters)
5      return {"data": results.transitions["S_to_I_total"]}
```

The simulation function serves as a wrapper around the `simulate` function, simply passing the arguments to it and returning a dictionary containing a key named `"data"`. This key stores the output quantity from the model which will be compared to the observed data using the distance function. It is important to emphasize that the simulation function must always return a dictionary containing the key `"data"`. The value associated with this key will then be passed to the distance function for the selection of stochastic trajectories during the calibration process.

As the second step, we define the prior distributions for the free parameters using the `scipy.stats` distribution functions:

```
1  from scipy.stats import uniform
2
3  priors = {
4      "beta": uniform(0.010, 0.020),
5      "mu": uniform(0.15, 0.1),
6  }
```

In both cases, we assume a uniform prior, with the transmissibility $\beta \sim U(0.01, 0.03)$ and the recovery rate $\mu \sim U(0.15, 0.25)$. It is important to note that in `scipy.stats.uniform`, the second argument does not specify the upper bound of the uniform range but rather the range's width.

Next, we define the `ABCSampler` object, which takes as inputs the simulation function, prior distributions, other parameters, the observed data, and the error metric (root mean squared error in this example):

```
1  from epydemix.calibration import ABCSampler, rmse
2
3  abc_sampler = ABCSampler(
4      simulation_function=simulate_wrapper,
5      priors=priors,
6      parameters=parameters,
7      observed_data=observed_data,
8      distance_function=rmse
9  )
```

We are now ready to run the calibration. To do so, we simply call the `calibrate` method of the `ABCSampler` object, specifying the calibration strategy and its specific arguments.

```
1  results_abc_rejection = abc_sampler.calibrate(
2      strategy="rejection",
3      num_particles=1000,
4      epsilon=550000
5  )
6
7  results_top_perc = abc_sampler.calibrate(
8      strategy="top_fraction",
9      Nsim=10000,
10     top_fraction=0.1
11 )
12
13 results_abc_smc = abc_sampler.calibrate(
14     strategy="smc",
15     num_particles=1000,
16     num_generations=5
17 )
```

In this example, we opted to run all three calibration algorithms. In particular, we use the `"rejection"` approach with 1,000 particles accepted and a tolerance of 550,000, the `"top_fraction"` approach with 10,000 total simulation

budget and top 10% selected, and finally the "smc" approach with 5 generations and 1,000 particles accepted in each generation.

Fig 3A compares the calibration results, plotting the observed data (black dots) alongside the median and 90% confidence intervals of the simulated new infections for each strategy. All three methods produce a good fit to the observed data, with slight differences in the confidence interval widths, reflecting variations in uncertainty across strategies. Fig 3B shows the joint posterior distributions of the transmission and recovery rates for each strategy, with the true parameters (red cross) used to generate the data. While all strategies converge on similar parameter regions, the density and shape of the posterior distributions differ, with the ABC-SMC resulting in the less disperse posterior. The similarity between ABC-SMC and ABC Rejection results is expected, as the tolerance used in ABC rejection matches the final generation's tolerance in ABC-SMC.

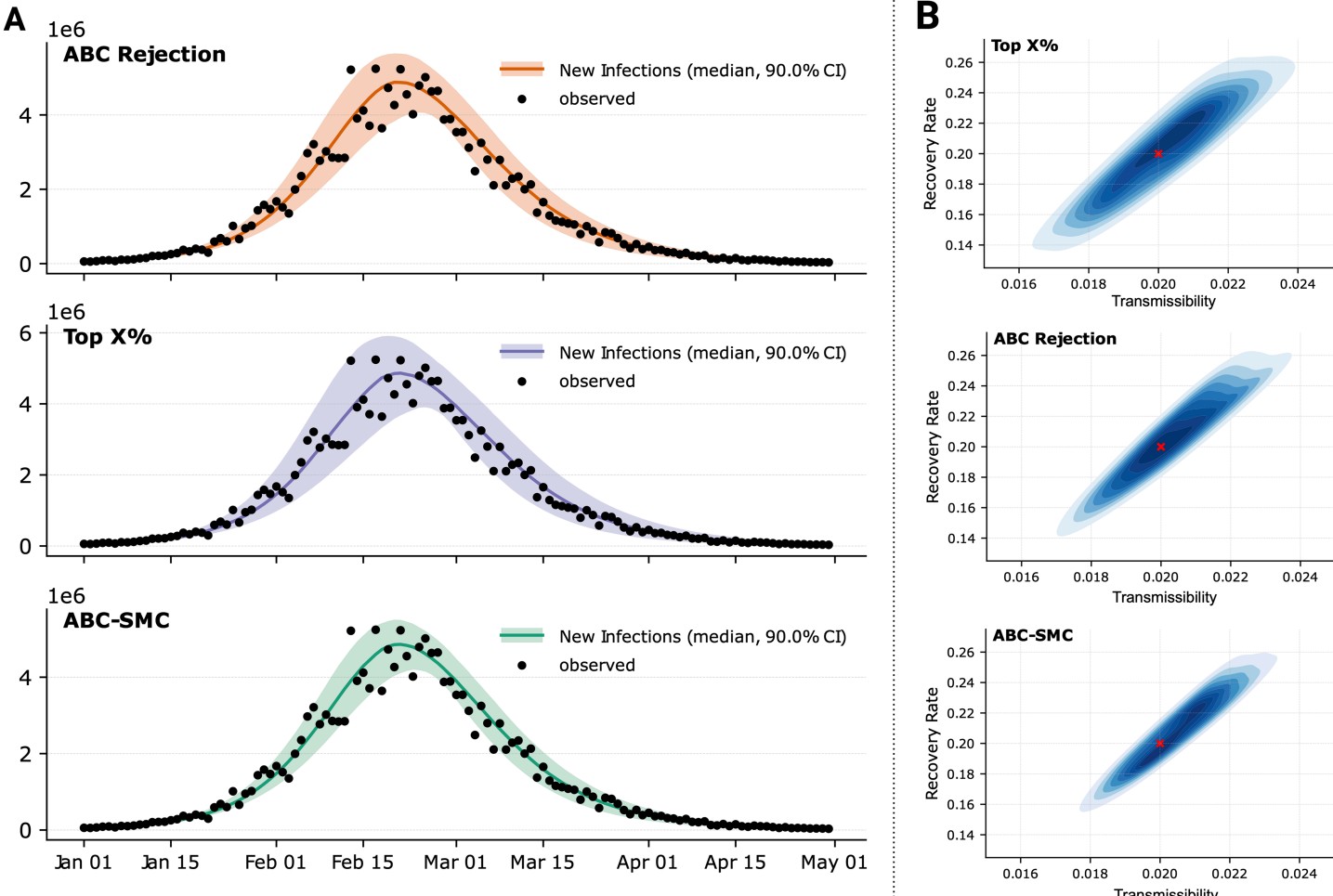

**Fig 3. A) Calibration results showing the fit of simulated new infections (medians and** 90% **confidence intervals) to synthetic data (black dots) for each calibration strategy. B)** Joint posterior distributions of the transmissibility and recovery rate for each strategy. The parameters used to generate the data are denoted with a red cross.

## 3.2 Modeling COVID-19 in Massachusetts

As detailed case study we develop a realistic epidemic model to reproduce the spread of SARS-CoV-2 in early 2020 in Massachusetts, USA. The model accounts for contact reductions due to social distancing recommendations and lockdowns, seasonal variation in transmissibility, and disease-related mortality. We calibrate the model's free parameters using real weekly reported deaths, between 2020/02/23 and 2020/06/01. The model accounts for 10 age groups (i.e., 0–9, 10–19, 20–24, 25–29, 30–39, ..., 80+) and incorporates population and contact matrices representative of the location under investigation. The compartmental structure of the model is illustrated in Fig 4A. Briefly, the model follows a SEIR-like structure with additional compartments to account for disease-related mortality. Specifically, multiple death-related compartments are included to represent the delay $\Delta$ of a few days between the end of the infectious period (i.e., exiting the $I$ compartment) and death (i.e., the final transition to $D_4$). This implementation allows considering Erlang distributions of delays [50]. In particular, we considered four intermediate transitions steps. The transmissibility $\beta$ is modulated by a seasonal factor $s(t)$, assumed to follow a sinusoidal pattern with an annual period, peaking in mid-January in the Northern hemisphere. The amplitude of this seasonal variation is treated as a free parameter. Additionally, the model incorporates a time-dependent modulation of contact rates, $r(t)$, estimated using mobility data from the COVID-19 Community Mobility Report published by Google [51]. Full details of the model, including the system of equations, are provided in the S1 Text. The code required to reproduce this analysis is more involved than the previous ones; therefore, we do not include snippets here. However, a full tutorial on this case study is available online on the package repository [35].

We calibrate the model using the ABC-SMC algorithm, with 10 generations and 1,000 particles per generation. Free model's parameters include the basic reproduction number $R_0$, the initial number of infected $I_0$, the amplitude of seasonal variation in transmissibility, and the delay $\Delta$ between the end of the infectious period and the time when deaths are reported. As detailed in the S1 Text, for all parameters we choose uniform flat prior distributions. In Fig 4B, we present the reported weekly COVID-19 deaths in Massachusetts alongside the median and 90% confidence intervals of simulated weekly deaths projected by the calibrated model. The projections closely align with the reported data, with a weighted median absolute percentage error of 18%. In the S1 Text we show the posterior distributions of free parameters. For $R_0$, the calibration yields a median of 2.18 with a 90% credible interval of [2.03,2.39].

After the calibration step, we perform out-of-sample scenario projections up to 2020/08/02, considering three different scenarios: (i) no relaxation of NPIs, where intervention levels remain unchanged from the last observed value in the calibration window, (ii) moderate relaxation, with contact rates increasing by 30%, and (iii) strong relaxation, with a 50%

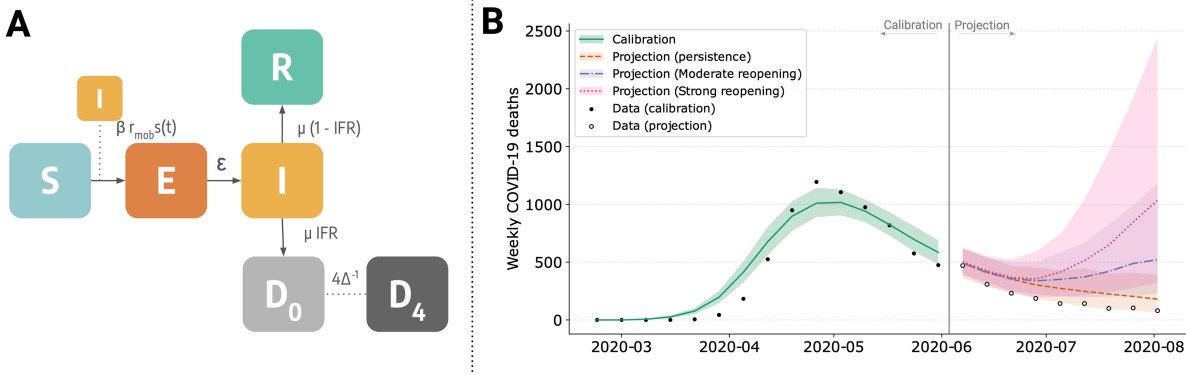

**Fig 4**. **A) Schematic representation of the extended SEIR model, incorporating additional compartments to account for disease-related mortality. B)** Projected weekly COVID-19 deaths in Massachusetts under different reopening scenarios. The black dots represent observed data, while the shaded areas indicate uncertainty intervals for the calibrated model (green) as well as different reopening strategies: status quo (orange), moderate (purple), and strong (pink).

increase in contact rates, approaching pre-pandemic levels. Additional details on the NPIs assumptions are provided in the S1 Text. In the first scenario, the downward trend in cases continues, aligning with the reported data. In the second, the trend reverses in mid-July, leading to a mild resurgence over the summer. In contrast, strong relaxation would have caused a significant resurgence in deaths during summer 2020.

## 4 Availability and future directions

In this work, we introduced Epydemix, a Python package designed to support the complete workflow of epidemic modeling, with particular emphasis on the often most challenging stage, namely model calibration.

Unlike many existing frameworks that provide only basic modeling components, Epydemix offers an integrated environment that spans the full modeling pipeline—from the definition of compartmental models incorporating socio-demographic structures for over 400 countries and regions, to the calibration of parameters using multiple Approximate Bayesian Computation techniques. Epydemix supports the implementation of both pharmaceutical and non-pharmaceutical public health interventions, providing freedom to define any number of compartments, ad-hoc transition types, and changes to both parameters and contact matrices as function of time. We developed a range of built-in methods to support the creation of visualizations to explore both inputs (e.g., properties of the contact matrices, population distributions) and outputs (e.g., incidence, or any other of target epidemic variable of interest). Furthermore, the package can also be used as a standalone calibration tool, allowing users to calibrate external models (i.e., developed without the Epydemix framework) based on data.

As with any Python package, Epydemix requires some initial familiarity with its design principles and usage patterns. To support users through this learning curve, we provide a series of detailed use cases—beyond those discussed in this work—available on the project's GitHub repository [34]. These examples complement the official documentation, which outlines the technical specifications and usage of the package's core classes and functions.

Looking ahead, several extensions are planned to broaden the scope and capabilities of Epydemix. Currently, we support only stochastic compartmental models. This choice was favored over deterministic (e.g., ODE-based) approaches which do not allow to capture stochastic fluctuations that are critical in small populations, near-threshold dynamics, and early outbreak phases. Nonetheless, we recognize that deterministic approaches can be advantageous in certain contexts, particularly when computational efficiency is a priority. Therefore, we plan to include deterministic simulation as a potential extension in future releases. Building on its current support for compartmental models, future releases will incorporate epidemic models based on explicit contact networks, followed by the integration of spatial modeling frameworks such as metapopulation structures. Performance enhancements are also a key priority, including the implementation of parallelization to accelerate simulation and calibration tasks. In addition, upcoming features will introduce tools and diagnostics to assess the practical identifiability of model parameters. These developments will further expand Epydemix's utility as a comprehensive and scalable platform for epidemic modeling.

Epydemix is designed to foster an active and collaborative community of users and contributors. To facilitate this, the package is released as open-source software under the GPL-3.0 license and is actively developed on GitHub [34]. The platform provides a space for users to report issues, engage in discussions, and contribute to the development of new features, helping us to ensure that the tool evolves in response to the needs of its user base. With its current capabilities and planned extensions, we believe that Epydemix is positioned to become a valuable resource for researchers and public health professionals, and to contribute to the reproducibility and accessibility of computational epidemic modeling tools.

## Supporting information

**S1 Text. In this supplementary material we present additional analyses and clarifications on the package implementation and usage.**
(PDF)

## Author contributions

**Conceptualization:** Nicolò Gozzi, Matteo Chinazzi, Jessica T. Davis, Corrado Gioannini, Luca Rossi, Nicola Perra, Alessandro Vespignani.

**Formal analysis:** Nicolò Gozzi.

**Methodology:** Nicolò Gozzi, Matteo Chinazzi, Jessica T. Davis, Corrado Gioannini, Luca Rossi, Marco Ajelli, Nicola Perra, Alessandro Vespignani.

**Software:** Nicolò Gozzi, Corrado Gioannini, Luca Rossi.

**Writing – original draft:** Nicolò Gozzi, Nicola Perra, Alessandro Vespignani.

**Writing – review & editing:** Nicolò Gozzi, Matteo Chinazzi, Jessica T. Davis, Corrado Gioannini, Luca Rossi, Marco Ajelli, Nicola Perra, Alessandro Vespignani.

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
