## [Decision Letter · Decision Letter 0]

15 Oct 2025

PCOMPBIOL-D-25-00960

Epydemix: An open-source Python package for epidemic modeling with integrated approximate Bayesian calibration

PLOS Computational Biology

Dear Dr. Gozzi,

Thank you for submitting your manuscript to PLOS Computational Biology. After careful consideration, we feel that it has merit but does not fully meet PLOS Computational Biology's publication criteria as it currently stands. Therefore, we invite you to submit a revised version of the manuscript that addresses the points raised during the review process.

Please submit your revised manuscript within 60 days Dec 15 2025 11:59PM. If you will need more time than this to complete your revisions, please reply to this message or contact the journal office at ploscompbiol@plos.org. Please include the following items when submitting your revised manuscript:

We look forward to receiving your revised manuscript.

Kind regards,

Eric Lofgren, MSPH, PhD

Academic Editor

PLOS Computational Biology

Denise Kühnert

Section Editor

PLOS Computational Biology

**Journal Requirements:**

2) Your manuscript is missing the following section: Availability and Future Directions. Please ensure that your article adheres to the standard Software article layout and order of Abstract, Introduction, Design and Implementation, Results, and Availability and Future Directions. For details on what each section should contain, see our Software article guidelines:

https://journals.plos.org/ploscompbiol/s/submission-guidelines#loc-software-submissions

1) If the funders had no role in your study, please state: "The funders had no role in study design, data collection and analysis, decision to publish, or preparation of the manuscript.".

6) Please ensure that the funders and grant numbers match between the Financial Disclosure field and the Funding Information tab in your submission form. Note that the funders must be provided in the same order in both places as well. Currently, "Lagrange Project of the ISI Foundation, funded by Fondazione CRT" is missing from the Funding Information tab.

7) Please revise your current Competing Interest statement to the standard "The authors have declared that no competing interests exist.". 

**Reviewers' comments:**

Reviewer's Responses to Questions

Reviewer #1: The paper presents a python package for the development and analysis of compartmental epidemic models, integrated with Bayesian parameter estimation methods. The package includes a number of features to ease model development, including assisted age-structure definition, time-varying parameters and time-dependent events, integration with real world demographic data.

I started reading the paper with a negative feeling. Developing a compartmental model from scratch in Python is not difficult, and also applying a parameter estimation method to it does not require too much work. However, while reading the paper, I completely change my mind. The proposed python package is not a basic API for model development, it is conceived in a way that simplify the work for the user and, at the same time, forces him/her to give a well-structured design to the model, favoring reusability and modification of the developed models by others. Apart from this, the package includes a number of features I found very interesting, such as the integration of real world demographic data. This is really nice: you develop a model and you want to test it on populations from different countries? You can do this easily. The integration of the parameter estimation method is, of course, another very useful feature. Finally, another aspect I liked of the paper, is the explicit reference to open-source development. Very often, research software is released with open-source licenses just to make it freely available to the community. In this case, the authors explicitly say that it is released with open-source license to stimulate community-based development. But, more importantly, it is by looking at the github repository that one understand that the aim is to open to external contributors (although for the moment only the first author committed there). Indeed, several commits have been done during the development (not only a final one, just to distribute the package), as for many OS projects there is a documentation on read-the-docs, and an issue opened by an external user and answered by the maintainer. These are small things but that show the real open-source attitude of the project.

Apart from all these positive aspects of the paper, there are also a number of things that should be improved. I list them below, mixing major and minor issues:

- Related work [MAJOR]. The description of related work is definitely insufficient. A lot of modelling approaches are listed and cited without a proper comparison. For example, many agent-based and network based approaches are cited, and I think these kinds of approach are nowadays particularly significant. Your framework instead is for the development of standard compartmental models. You should motivate your development choice and the fact that you plan to extend in the direction of metapopulation models (as said in the final discussion). Moreover, the description of related work about parameter estimation/model calibration is mostly missing. It consists of a single paragraph ("As mentioned, ...") with one only citation. You should improve significantly this part. During covid-19 a lot of papers have been published in which parameter estimation methods are applied to sir models: you should review them and compare the Bayesian approaches you adopted with those in those papers.

- Line 147, I think "X = [S,I,R]" should be replaced by "X \in \{S,I,R\}"

- Line 155. Your library works only with stochastic models. I understand that stochastic models can be preferable for several reasons, but several times ODE based models could be better (at least for performance reasons). So, forcing stochastic modelling could be seen as a limitation. You should discuss this choice and possibly consider to include ODE based simulation to be included in a future version of the package.

- Lines 162-165. You should be more precise in describing these rates, possibly by formalizing their definition. In the case of the spontaneous transition, for example, it seems that the parameter provided by the modeler is directly used as rate (as is) while typically the rate is computed by multiplying the parameter by the size of the compartment...

- Whole section 3 [MAJOR]: the whole description of the design and implementation is in abstract terms. Understanding would be made easier by the addition of examples throughout the text. Please, revise the whole section, trying to make it easier to read for the non-deeply-expert reader.

- Line 170. "simulation parameters" Which parameters? not clear

- Section 3.3.3. Are all these interventions and changes time dependent? (if not, clarify) It would be very useful to include changes that are condition-based, namely that are triggered when a given condition on the simulation state is satisfied (e.g., lockdown when ratio of infected is >10%). This would be very useful, I think.

Reviewer #2: The Article introduces Epydemix, an open-source Python package for the development and

calibration of stochastic compartmental epidemic models. Contrary to previous works, it also integrates Approximate Bayesian Computation (ABC) to infer model parameters on observed data. The authors present various usage examples displaying the flexibility of their framework.

The article is extremely well written and enables professionals and researchers to easily employ the software. The examples provided are clearly described and effectively showcase the potential of the framework. The theoretical assumptions underlying the epidemic modeling are explicitly stated and fully consistent with the objectives of the work. I believe this publication will greatly benefit the entire community.

**Have the authors made all data and (if applicable) computational code underlying the findings in their manuscript fully available?**

Reviewer #1: Yes

Reviewer #2: Yes

PLOS authors have the option to publish the peer review history of their article (what does this mean?). If published, this will include your full peer review and any attached files.

Reviewer #1: **Yes: **Paolo Milazzo

Reviewer #2: No

**Figure resubmission:**
---

## [Editor Report · Decision Letter 1]

12 Nov 2025

Dear Gozzi,

We are pleased to inform you that your manuscript 'Epydemix: An open-source Python package for epidemic modeling with integrated approximate Bayesian calibration' has been provisionally accepted for publication in PLOS Computational Biology.

Best regards,

Eric Lofgren, MSPH, PhD

Academic Editor

PLOS Computational Biology

Denise Kühnert

Section Editor

PLOS Computational Biology

---

## [Editor Report · Acceptance letter]

PCOMPBIOL-D-25-00960R1

Epydemix: An open-source Python package for epidemic modeling with integrated approximate Bayesian calibration

Dear Dr Gozzi,

I am pleased to inform you that your manuscript has been formally accepted for publication in PLOS Computational Biology. Your manuscript is now with our production department and you will be notified of the publication date in due course.

With kind regards,

Anita Estes
